# Leveraging Exosomes as the Next-Generation Bio-Shuttles: The Next Biggest Approach against Th17 Cell Catastrophe

**DOI:** 10.3390/ijms24087647

**Published:** 2023-04-21

**Authors:** Snigdha Samarpita, Xiaogang Li

**Affiliations:** 1Department of Internal Medicine, Mayo Clinic, Rochester, MN 55905, USA; 2Department of Biochemistry and Molecular Biology, Mayo Clinic, Rochester, MN 55905, USA

**Keywords:** exosome engineering, Th17 cell, drug delivery vector, packaging therapeutics

## Abstract

In recent years, the launch of clinical-grade exosomes is rising expeditiously, as they represent a new powerful approach for the delivery of advanced therapies and for diagnostic purposes for various diseases. Exosomes are membrane-bound extracellular vesicles that can act as biological messengers between cells, in the context of health and disease. In comparison to several lab-based drug carriers, exosome exhibits high stability, accommodates diverse cargo loads, elicits low immunogenicity and toxicity, and therefore manifests tremendous perspectives in the development of therapeutics. The efforts made to spur exosomes in drugging the untreatable targets are encouraging. Currently, T helper (Th) 17 cells are considered the most prominent factor in the establishment of autoimmunity and several genetic disorders. Current reports have indicated the importance of targeting the development of Th17 cells and the secretion of its paracrine molecule, interleukin (IL)-17. However, the present-day targeted approaches exhibit drawbacks, such as high cost of production, rapid transformation, poor bioavailability, and importantly, causing opportunistic infections that ultimately hamper their clinical applications. To overcome this hurdle, the potential use of exosomes as vectors seem to be a promising approach for Th17 cell-targeted therapies. With this standpoint, this review discusses this new concept by providing a snapshot of exosome biogenesis, summarizes the current clinical trials of exosomes in several diseases, analyzes the prospect of exosomes as an established drug carrier and delineates the present challenges, with an emphasis on their practical applications in targeting Th17 cells in diseases. We further decode the possible future scope of exosome bioengineering for targeted drug delivery against Th17 cells and its catastrophe.

## 1. Introduction

The advancement of novel computational tools and approaches, such as computer-aided drug design, high-throughput screening and artificial intelligence, helps expedite drug discovery and identify an active pharmaceutical ingredient (API) as new lead compounds in several disease settings [1]. Howbeit, several drugs escape from reaching the target sites due to an individual’s biological barrier and immune defense mechanisms during the process of disease treatment. Moreover, unwanted toxicities due to their accumulation in off-target tissues have been a major bottleneck in their translation into clinics [2,3]. Thereupon, strategies that enhance tissue-specific drug delivery hold promise to overcome these limitations and demonstrate tremendous success in parallel with drug discovery.

Exosomes are exemplified as “membrane-anchored extracellular vesicles” derived from the endosomal compartment of almost all eukaryotic cells [4]. As a nanoscale-sized vesicle, exosomes are found in all biological fluids (urine, blood, saliva, semen, breast milk and cerebrospinal fluid), which fabricate them as an excellent biomarker in the diagnosis of cancer and related disorders [5]. At the forefront, the exosome adapts a specific mechanism that includes surface receptor-mediated endocytosis, pinocytosis and membrane fusion to transport cargo to recipient cells [6]. As exosomes are imprinted with the ability to facilitate cell–cell communication, it is an unbiased claim that they represent potent drug delivery vectors in clinics due to their characteristic properties of innate stability, low immunogenicity, negative zeta potential to escape the immune attack and good capacity to penetrate through biological barriers [7].

Exosomes are extensively explored as tools in the landscape of immune regulation and immunotherapies, harnessing their role in immune surveillance, antigen presentation, immunosuppression, and anti-tumor immunity [8]. Proteomic studies have demonstrated that exosomes are tagged with immune function-related proteins, such as human leukocyte antigen (HLA)-1, β2-microglobulin and sub-units of the T cell receptor (TCR)-CD3 complex, thus defining their essential involvement in immune regulation [9]. For instance, cancer cell-derived exosomes activate vascular endothelial growth factor (VEGF) signaling in endothelial cells and advance angiogenesis in the tumor microenvironment [10]. In addition, exosomes exhibit immunosuppressive function via inhibition of CD8^+^ T cells, programmed death ligand 1 (PD-L1) secretion and expansion of suppressive regulatory T cells [11,12]. Notwithstanding, exosomes in the synovial fluid encapsulate citrullinated proteins that promote the formation of auto-immunogenic determinants [13]. However, dendritic cells (DCs)-derived exosomes express major histocompatibility complex class (MHC)-I molecules, which contribute to antigen presentation and anti-tumor responses [14], making them a hopeful prospect from the therapeutic standpoint of exosomes. Thereupon, the engineering of exosomes to deliver peptides, vaccines or drugs into specific cells in a time-dependent manner can be a star therapeutic strategy to negotiate immune responses with regard to immune-related diseases.

In various diseases such as cancer, autoimmune diseases and most genetic disorders, there exists an intricate imbalance between the occurrence of inflammatory events and the suppression of inflammation, which instigates disease progression [15]. Of note, immune cells are important modulators that facilitate this imbalance. In this context, an increased frequency of Th17 cells, a subset of CD4^+^ T immune cells, has been studied to promote the initiation and development of several tumors and autoimmune/inflammatory disorders via distinct mechanisms [16,17]. Thus far, targeting Th17 cells is an unmet clinical need. At present, exosome-based immune therapy is a jackpot as an effective strategy to restore immune balance and halt disease pathogenesis. In cancer, for example, curcumin export by exosomes facilitates the differentiation of effector T cells to kill cancer cells [18]. Under this notion, there are a few discussions on how exosomes communicate and can target Th17 CD4^+^ T cells to impede the development of life-threatening diseases.

Will exosomes be a solid answer to this ongoing search for a promising Th17-targeted drug delivery platform in clinics? With this explorative question in mind, this review will present an overview of the prospects of exosomes as a delivery platform for Th17-targeted therapies. As numerous reviews are published on exosomes, our initial discussion will be generalized to contextualize the growing importance of exosomes as “postmasters” for drug delivery. However, this review will be the first to provide a layout of exosomal payloads that repress Th17 cell function. We will update our readers on the clinical trials of exosomes, with an emphasis on the Th17 paradigm and current practical challenges faced in the race. Addressing these challenges will provide a roadmap to figure out exosome engineering strategies and categorize them based on their benefit scores to best exploit them for Th17 cell-targeted therapy. Finally, we will highlight the prospects of incorporating different small molecules, peptides, vaccines, antibodies and genetic moieties as exogenous payloads into exosomes to maximize the desired therapeutic applications against Th17 cell-mediated diseases. This review is contemplated to set a significant benchmark for designing the exosomes for successful drug encapsulation and render them as a facile and promising approach in the field of Th17 cell-targeted therapy in clinics.

## 2. The Structure of Exosomes and Its Composition

Exosomes are defined as lipid-bilayered extracellular vesicles with a diameter ranging from 30 to 200 nm. Secreted naturally from almost all kinds of cells, these vesicles are formed through endosomal membrane budding and constitute the nucleic acids, proteins, carbohydrates and lipids of a cell [19]. Moreover, these nanosized biological vectors shuttle cell-derived nucleic acids, lipids and proteins as signaling molecules to facilitate signal transduction and communication between neighboring cells [19]. Previous researchers have demonstrated that exosomes are closely associated with viral responses, immune regulation, cardiovascular risks, central nervous system (CNS) disorders, pregnancy, cancer and autoimmune disease progression [4]. Numerous efforts are in progress to isolate circulating exosomes that import biological substances and dynamically expand them as biomarkers as they can manifest several pathophysiological conditions [20]. Emphatically, exosomes have been explored as drug delivery vectors due to their ability to inherit characteristic properties of a cell and exhibit a better safety profile as compared to cell-based therapy [21]. Instead, molecular insights to understand the structures of exosomes could be helpful to engineer exosomes for loading cargoes to ferry them into target cells/tissues.

The contents inside and on the surface of the exosomes mirror the configuration of the parent cell. However, irrespective of their parental origin, some features including tetraspanins (CD9, CD81, CD82 and CD63), syndecans, heat shock proteins (Hsp60, Hsp20, Hsp70, Hsp22, Hsp90 and alpha-B Crystallin), biogenesis-associated proteins (ESCRT proteins, CHMP4, TSG101, STAM1, VPS4, PLD2), membrane transport and fusion proteins (annexins, GTPases, and Rab molecules), nuclear acids (long non-coding RNAs, miRNA, mRNA and DNA) and lipids, commonly span the membrane of exosomes structure [22]. Tetraspanins are abundant on the exosomal membranes to form a complex with integrins and contribute to the tropism of exosomes [23]. Advanced proteomic studies revealed that heat shock, transport and fusion proteins mediate the sprouting of multicellular vesicular bodies (MVBs) [24], whereas the lipidomic data demonstrated that the lipid contents of exosomes are either conserved or relatively similar and are crucially involved in preserving the shape of exosomes and maintaining homeostasis in the recipient cells [25]. Moreover, the asymmetrical distribution of the lipid bilayer of exosomes allows for successful drug delivery. For example, exosomes that display α_6_β_4_ integrins are likely to target S100-A_4_ positive fibroblasts and surfactant protein C (SPC)-positive epithelial cells of the lungs [26]. Similarly, lymphocyte function-associated antigen 1 (LFA-1)-tagged exosomes act as a “molecular Trojan horse” that can cross the blood–brain barrier to treat central nervous system (CNS)-related disorders [27], and αvβ6 exosomes play a critical role in interstitial inflammation and fibrosis [28].

The membrane of the exosomes is composed on the basis of their origin and physiological environment during exosome biogenesis [4]. Accordingly, the target site depends on the differential expression of markers on the surface of exosomes [29]. As a proof of concept, exosomes derived from dendritic cells express MHC class I, MHC class II, ICAM, CD40 and CD86 [30]. Also, chimeric antigen receptor T cell immunotherapy (CAR-T) cell-based exosomes express CARs and contain cytotoxic molecules used for cancer therapy [31], and those released by natural killer (NK) cells express lymphocyte function-associated antigen 1 (LFA-1) and DNAX accessory molecule 1 (DNAM-1) [32], thus, mirroring the characteristic features of parent cells. Exosomes generated from γδ T cells carry death-inducing ligands (FasL and TRAIL) and CD80/86 and MHC class I/II with dual tumor-killing effects [33]. Information on the membrane structures of exosomes that can guide their destination is available in different web-based sources, including the International Society for Extracellular Vesicles, American Society for Exosomes and Microvesicles, Extracellular RNA Communication Program, ExoCarta, Vesiclepedia and EVpedia.

## 3. Mechanistic Insights into Exosome Biogenesis and Release

The biogenesis of exosomes begins with the formation of intraluminal vesicles (ILVs) that develop via inward budding of the plasma membrane (early endosomes), which then generate late endosomes composed of combined intraluminal vesicles (ILVs), in a clathrin- or caveolin-dependent or -independent manner, called multivesicular bodies (MVBs) [19]. Based on the pathway they follow, they are categorized as degradative multivesicular exosomes (MVE) which coalesce with the lysosomes to ensure degradation of intraluminal contents and secretory MVE, which pinches out of the membrane to discharge its contents into the extracellular space. Moreover, MVE also leads to the formation of endosomal-related organelle, melanosomes and Weibel–Palade bodies in pigment and endothelial cells, respectively.

The biogenesis and protein sorting of ILVs are tightly regulated processes requiring an endosomal sorting complex (ESCRT) or can be an independent process [19]. In the context of the ESCRT-dependent pathway, it is composed of ESCRT-0, ESCRT-I, ESCRT-II and ESCRT-III complexes and AAA ATPase VPS4, tumor susceptibility gene (Tsg101) and ALIX auxiliary proteins [34]. EXCRT-0 escorts mono-ubiquitinated proteins into the endosomal domain with the help of cytosolic protein hepatocyte responsive serum phosphoprotein (HRS) heterodimer, signal transducing adapter molecule (STAM)1/2, Eps15 and clathrin. In the next sequential step, ESCRT-I and ESCRT-II together with ESCRT-0 bind to ubiquitinated cargo with higher affinity to form a stable membrane neck. Finally, ESCRT-III assembles the complex for the excision of the membrane neck and internalization of buds into the endosomes. Eventually, the cargoes are de-ubiquitinated by de-ubiquitylating enzymes and, ATPase VPS4 dissociates and recycles the endosomal components for the next cycle [34]. However, if de-ubiquitination is restrained, ILVs are subjected to lysosomal degradation [35]. Alternatively, the marker protein of exosomes, Alix, binds to ESCRT-III and guides the un-ubiquitinated cargo to the ILVs [34,35]. Besides it, the ESCRT-independent mechanism occurs via the melanosomal protein Pmel17 in melanocytes [36]. The association of the luminal domains of Pmel17 along with lipids contributes to ILV formation. It has also been reported that tetraspanin mediates cargo sorting and exosome secretion in an ESCRT-independent manner [37]. For instance, tetraspanin CD63 helps the loading of pigment cell-specific protein 17 into the ILV during the synthesis of the melanosome [37]. Similarly, the loading of MHC-II occurs exclusively in tetraspanin CD9-enriched membrane domains [38]. In addition, as reported previously, ceramide-rich parts of endosomes are prone to inward invagination, and hence, defects in the conversion of sphingomyelin into ceramide by sphingomyelinase (SMase) abrogates ILV formation, wherein, cholesterol is necessary for the formation of curved membrane structures called caveolae [39]. This has been confirmed for lipid-based cargo sorting as well as exosome secretion. It is imperative to understand that the absence of ESCRT machinery did not hamper the MVB formation, but instead manipulated ILV number and size, thus exemplifying coordination between ESCRT-dependent and -independent pathways during exosome biogenesis.

In view of exosome release, the involvement of molecular switches and motors (small GTPase, dynein and kinesin), cytoskeletal elements (microtubule and microfilaments) and membrane fusion complexes (SNARE complex) has been made crystal clear [40]. Rab GTPases control the trafficking and fusion of MVB with the plasma membrane. During the transport of MVB, the microtubule cytoskeleton and molecular motors direct its polarized distribution in the cells [40]. Rightly, the soluble N-ethylmaleimide (NEM)-sensitive factor attachment protein receptor (SNARE) complex is a core protein complex that assists membrane fusion and secretion [41]. Descriptively, the fusion process is initiated through the interaction of the plasma membrane protein syntaxin with synaptotagmin, the calcium sensor located on MVBs. Then, v-SNAREs of MVBs pair with the t-SNAREs, thus driving SNARE complex formation and fusion of membranes to release exosomes to the outer environment [40,41] (Figure 1). Further understanding of the complex process of exosome biogenesis should pave the way for novel therapeutic avenues.

With much detailed apprehension on the biogenesis of exosomes, efforts are in progress to inhibit the production or release of an exosome subpopulation that is involved in the pathology. Nevertheless, the available inhibitors are listed in Figure 1.

## 4. Exosome Classification and Its Biodistribution: Current State-of-the-Art

At least two distinct sub-types of exosomes have been discovered based on their modifications—natural exosomes and bioengineered exosomes. Subsequently, natural exosomes are subdivided into animal-derived exosomes and plant-derived exosomes based on their parental sources. With reference to animal-derived exosomes, almost all normal cells such as mesenchymal stem cells (MSCs), macrophages, natural killer (NK) cells, T and B immune cells and umbilical cord endothelial cells produce exosomes. Also, normal exosomes are plentily available in biofluids, such as plasma, urine, milk and saliva. In essence, tumor-cell-derived exosomes, because of their specific expression of tetraspanins, have also been considered for the delivery of chemotherapeutic or anti-cancer drugs. However, tetraspanins can sometimes lead to tumor growth and an increase in metastasis in certain types of cancer [42]. Therefore, the alarming risk associated with tumor exosomes jeopardizes the suitable therapeutic effects and aggravates patients’ malignancies, thus providing an awakening call to weigh the benefit-to-risk ratio and choose the right exosomes for therapy purposes. Based on these perspectives, several plant-derived exosomes have been explored for clinical use, as they are derived from purified sources and are considerably safe [43]. Contrastingly, engineered exosomes fall under the category of exosomes that are surface modified and loaded with substances to efficiently target disease sites and negate adverse profiles associated with treatments. With no available information to date, further sub-divisions of exosomes based on organophilicity, biodistribution and clearance may be considered [42,43].

Irrespective of the parental origin, it has been stated that the exosomes are largely accumulated in the liver, spleen, kidney and lungs. However, there is evidence that states that exosomes have asymmetric biodistribution based on their parental cell and the route of administration. Like, glioma-derived exosomes are demonstrated to efficiently deliver the drug to the brain [44]. Mesenchymal stem cell (MSC)-derived exosomes preferentially target kidneys in patients with kidney disorders [45]. Besides the exosome origin being one of the striking reasons, the route of administration can also be equally considered to alter exosome biodistribution. For instance, intramuscular administration of exosomes increases their bioavailability in the gastrointestinal tract. Alternatively, the oral gavage method of administration localized it more in the intestine. However, the most common method of administration is intravenous, in which exosomes travel via the heart and are entrapped in the capillaries of the lungs [46]. This highlights that the method of administration can lead to extreme variability and an asymmetric biodistribution pattern for exosomes.

## 5. Exosome in Diseases: The Boon against Evil

Capitalizing on the role of exosomes in the progression or treatment of several diseases is an important aspect of this review. Earlier described as “waste disposal units”, exosomes are now a hot research topic as they proportionately function between physiological and several pathological states. Finding how exosomes communicate with or affect neighboring/distant cells may provide insights into their roles in disease regulation, exploit them for drug delivery or monitor disease state in a non-invasive procedure.

Investigating the darker side of the exosome reveals that it secretes β-amyloid and hyperphosphorylated, misfolded tau into the extracellular space and leads to the pathological spreading of tauopathy in Alzheimer’s patients [47]. Exosomal alpha-synuclein (α-syn) promotes the aggregation of misfolded α-syn, a phenotype predominant in Parkinson’s disease [48]. An increase in the levels of circulating exosomes loaded with pro-inflammatory cytokines leads to persistent inflammation, as has been reported in multiple sclerosis and autoimmune encephalomyelitis [49]. In autoimmune rheumatoid arthritis (RA), exosomes derived from synovial fibroblasts and serum-induced Th17 differentiation and production of pro-inflammatory cytokines aggravate disease conditions [50]. In genetic diseases such as autosomal dominant polycystic kidney disease (ADPKD), cystic cells and urinary exosomes promote cyst growth in ADPKD patients [51]. In addition, fibroblasts of breast cancer origin secrete exosomes that mediate epithelial-to-mesenchymal transition (EMT) and metastasis via altered expression of lncRNA. A detailed analysis of miRNAs, lncRNAs, circRNAs and proteins that promote several diseases has been categorized in Table 1.

Owing to their ability to deliver mRNA and proteins to recipient cells at distant sites, exosomes can promote tumor metastasis and exert immune suppression to evade attack by the immune microenvironment. Gastric cancer cell exosomes establish an immunosuppressive environment for metastatic niche formation and exacerbate lung tumor metastasis [66]. In type 2 diabetes, exosomes released from adipocytes carry thrombospondin 5, which induces metastatic properties associated with the mesenchymal phenotype of breast cancer cells [67]. It is important to note that exosomal PD-L1 exhibits immunosuppressive function in the tumor microenvironment and promotes resistance to immune checkpoint blockade [12]. To note, exosomal PD-L1 from MEL624 cells decreased the proliferation of CD8^+^ T cells and reduced granzyme B secretion in an in vitro model of melanoma [68]. Moreover, tumor-derived exosomes re-educated neutrophils to exhibit an immunosuppressive microenvironment to promote gastric cancer via high mobility group box-1 (HMGB1) activated STAT-3 expression and increase PD-L1 activity [69]. Exosomes derived from patients with non-small cell lung cancer expressed PD-L1 and exhibited tumor escape via decreased IL-2 and IFN-γ secretion [70]. In a syngeneic model of prostate cancer, exosomal PD-L1 was resistant to anti-PD-L1 therapy, and its genetic blockade improved systemic anti-tumor response, strongly indicating the disease-promoting effect of exosomal PD-L1 towards tumor progression [71].

In another instance, exosomes shed from colon cancer cells showcased cetuximab resistance via inhibiting PTEN expression and concurrently, increasing the functional activity of the Akt pathway [72]. Interestingly, exosomes release circular RNA (circSKA3) that potentiates the formation of large colonies and the maintenance of invasiveness in breast cancer [73]. As evident, exosomes from the serum of patients with prostate cancer transfer pyruvate kinase M2 into bone stromal cells and promote its metastasis [74]. Exosome RNF126 derived from tumor cells promoted PTEN ubiquitination and macrophage infiltration, which led to nasopharyngeal carcinoma progression [75]. Melonoma-based exosomes increased PD-1 expression and reprogrammed mesenchymal stem cells to activate oncogenic and survival signals toward tumor metastasis [76]. Tumor exosomes display Tspan8 and CD151 that stimulate angiogenesis and tumor progression [77]. Cancer cell-derived exosomes are reported to stabilize cadherins and promote epithelial to mesenchymal transition that progresses cancer [78]. Taken together, the accumulating evidence concludes the cancer-promoting function of exosomes by subverting immune regulation.

As a bystander in therapy, exosomes of different origins have intrinsic remedial properties that can treat diseases, including cardiovascular, respiratory, kidney and brain disorders. Exosomes have the ability to increase drug half-life and stability by protecting them from degradation by digestive enzymes, and hence, serve as better vehicles for drug delivery in the treatment of cancer and other related diseases. Owing to the numerous undergoing clinical trials of exosome-based therapy, it can be speculated that it might contribute to better responses in patients and can thus be included under standard-of-care therapy for several diseases in the near future (Table 2) [79,80,81,82]. The incorporation of paclitaxel into exosomes exhibited better therapeutic efficacy in pancreatic cancer cells and in NODscid mice (*n* = 12) [83]. Exosomes have also been used to increase the stability and bioavailability of curcumin to mitigate inflammation and brain tumor growth (*n* = 5 mice per group) [84]. In fact, curcumin pre-treated lung cancer cells increased exosomal transcriptional factor 21 (TCF21) to subside lung cancer [85]. Doxorubicin-loaded exosomes exhibited higher efficacy with minimal levels of cardiotoxicity [86]. Similarly, exosomes released withaferin at target sites have been tested to treat lung tumor growth [87]. Besides transporting natural cargoes, exosomes are best suited as nanocarriers for the delivery of RNA (mRNA, siRNA, miRNA)-based therapeutics (Table 2). Markedly, specific disease-associated proteins and mRNAs of exosomes can be used as diagnostic markers. Apart from its role in delivering drugs and diagnosis, exosomes can be labeled with CFSE dyes, GFP-expressing plasmids and lipophilic agents (PKH67, Di dyes) to monitor treatment efficacy [88]. In addition, the bioluminescence resonance energy transfer (BRET) imaging technique scans the homing of exosomes at the target tissues and organs, which can be used to guide the development of therapeutics [89]. As a therapeutic carrier, diagnostic tool and best imaging agent, exosomes as a comrade outshine their criminal story and prove to be a boon in the medical field.

## 6. T Cell Exosomes: Its Biogenesis and Applicability

Although the information about the biogenesis of exosomes from T cells is preliminary, the omnipresent expression of tetraspanins makes it amenable to be involved in T cell exosome biogenesis [90]. Certain lipid metabolizing enzymes are also plausible to promote the maturation and release of exosomes by T cells. For instance, diacylglycerol kinase α helps in the secretion of CD63 exosomes from T cells [91]. An important clue for the involvement of condensed membranes in exosome formation is derived from the fact that exosomes derived from T cells are enriched with sphingomyelin and cholesterol. In addition, the participation of Tsg101 and, reportedly, HRS (exosomal markers) in exosome biogenesis rules out ESCRT components in the T cell biogenesis process [90].

## 7. Exosome Shedding at the Crossroads of Immune Synapse

It has been postulated that mature immune synapse formation launches exosome release from T cells. In this context, T-cell receptor (TCR) activation on the effector T cell induces the secretion of exosomes bearing CD63 and TCR, wherein different subpopulations of exosomes are released with the presence/absence of T cell co-stimulatory molecule signaling [92]. During the synapse formation, the secretory vesicles are polarized towards the central region of the formed immune synapse, called the central supramolecular activation cluster (cSMAC) for fusion and subsequent secretion in the synaptic cleft [93]. Disruption of Tsg101 and VPS4 interrupts the sorting of TCR into the secretory vesicle and secretion respectively, thus, rendering the important role of ESCRT components (Tsg101 and VPS4) in T cell exosome biology [92,93]. Thus, exosome synthesis in T cells upon encountering an antigen is a unique approach, and several tetraspanins, lipid components and ESCRT machinery can be pharmacologically targeted to block exosome secretion by T cells.

Regarding the functional role of T cell-based exosomes, it has been evidenced that exosomes with Fas ligand derived from T cells are lethal and induce death in other T lymphocytes [94]. It is also observed to promote tumor invasion in the lungs via the secretion of matrix metalloproteinase (MMP)-9 [90]. CD3^+^ T cell-derived exosomes function in accordance with IL-2 and promote the multiplication of autologous resting cells [95]. In addition, CD73-tagged exosomes released from T regulatory (Treg) cells induce the production of adenosine, which exhibits an anti-inflammatory response [96]. Furthermore, Treg cell-based exosomes release let-7b/7d and miRNA-I55 and abrogate Th1 differentiation and interferon (IFN)-γ production [97], and secretion of miR-150-5p and miR-142-3p to DCs suppresses TNF-α and IL-6 production via the increased amount of IL-10 cytokine production [98]. The levels of inhibitory molecules PD-1 and TIM-3 are increased in normal CD8 T cells as a result of lncRNA KCNQ1OT1-loaded T cell exosomes [99]. Yet importantly, CAR-T cell exosomes could surpass the disadvantages of CAR-T cell therapy and tend to dominate as therapeutic agents [98].

## 8. Th17 Cell Biology

CD4^+^ T cells are the central tenet in the promotion and maintenance of immune response. Defined as helper cells with a heterogeneous population, CD4^+^ T cells are categorized based on the transcription factor and cytokine profile. T helper (Th)-17 cells are a distinct subset of CD4^+^ T cells that expresses lineage-defining transcription factor RAR-related orphan receptor gamma (RORγT) and secretes IL-17 cytokine [100]. The differentiation and commitment of CD4^+^ Th17 cells is an intricate process and involves the “non-cytokine” mechanisms and “cytokine-induced” transcriptional program. The non-cytokine processes include the nature of antigen-presenting cells and their microbial peptides, the strength of interaction between TCR and antigenic peptide on MHC molecules and the activation of co-stimulatory signals [101]. It has long been reported that antigens from the fungus *Candida albicans* and its associated PAMP β-glucan differentiate fungus-specific Th17 cells [102]. However, there is a lack of concrete evidence regarding the TCR signals toward Th17 cell commitment. It is of the opinion that weak TCR signals in the proximity of Th17-promoting cytokines curb Th17 cell differentiation [103]. But, strong TCR signaling mediated activation of Tec family tyrosine kinase, interleukin-2-inducible T cell kinase (Itk), that skews the differentiation of CD4 T cells to Th17 phenotype. Interestingly, the binding of the nuclear factor of activated T cells (NFATc1) to the promoter region of IL-17 is also essential to differentiate Th17 cells even in the presence of strong TCR signals [104]. Alongside, low CD28 and high CD40-CD40L co-stimulatory signals with impaired IL-2 levels strengthen the Th17 developmental axis [105]. In sum, the nature of the antigen and the strength of TCR and co-stimulatory signals are instrumental in the biased commitment of CD4^+^ T cells to a distinct Th phenotype.

Along several lines, the inclusion of several cytokines and transcription factors (TFs) have been well reported that guide Th17 differentiation and lineage commitment. Particularly, TGF-β and IL-6 are the important cytokines that prime Th17 differentiation [106]. Furthermore, IL-21, IL-23 and IL-1β contribute to Th17 cell phenotype commitment [107]. Likewise, some TFs also have been demonstrated to participate to regulate the Th17 development process. Notably, in response to IL-6 cytokine stimulation, the STAT-3/ROR-γT axis is activated that mediates the formation of Th17 cells and the IL-17 transcriptional program [107]. Unlike ROR-γT, IL-23 promotes AhR transcription factor activation to expand the Th17 population [108]. It is likely that the induction of Th17 cell plasticity is also governed by interferon regulatory factor 4 (IRF4) TF, and IRF4 deficient cells impair IL-17 production and Th17 responses under Th17 differentiation conditions [109]. Furthermore, another transcription factor basic leucine zipper ATF-like transcription factor (BATF) forms a complex with IRF4 on the IL-17 promoter region downstream of TCR signaling [110]. However, the independent function of these transcription factors is yet to be determined along the Th17 differentiation axis. Though not much explored, the importance of Ying Yang 1 (YY1) has been demonstrated to bind to the T-bet promoter region and influence Th17 pathogenicity [111]. Integrated genome studies have further revealed that Zinc finger E-box binding homeobox 1 (ZEB1) necessitates JAK-2/STAT-3 dependent Th17 cell fate [112].

In addition, RNA-binding proteins also control Th17 differentiation and function. Reportedly, human antigen R (HuR) post-transcriptionally promotes ROR-γT expression and Th17 cell responses [113]. Very recently, the role of IGF2 mRNA-binding protein 2 (IMP2) RNA-binding protein drives the Th17 cell differentiation program and infiltration into the lymph nodes of the inflamed brain [114]. In sum, a detailed understanding of the factors that regulate Th17 biology will help open new avenues for exosome-targeted therapies in Th17-mediated diseases (Figure 2).

## 9. Th17 Cell Pathologies: A Snap on Th17 Exosomes

The pace in the use of IL-17 and ROR-γT inhibitors in clinics has resurfaced the critical role of Th17 in the exacerbation of several cancers, autoimmune diseases, and genetic disorders. Recently, Th17 cell populations are aggressively increased in COVID-19 patients with pulmonary complications. There are reports that correlate Th17 with neutrophilia and NETosis pathologies in COVID patients. Anti-IL-17 monoclonal antibodies (mAbs) netakimab administered to COVID patients mitigated lung lesions and demand for oxygen support [115]. Also, influenza virus-induced lung injury was associated with increased levels of IL-17 and Th17 population [116]. IL-17 contributes to liver fibrosis and necrosis of hepatocytes in mice infected with the herpes virus [117]. Notwithstanding, stromal keratitis in mice with corneal infection is an outcome of Th17 cells [118]. Furthermore, Th17 cells mediate myocarditis in coxsackievirus B3-infected mice [119], suggesting that targeting Th17 cells and IL-17 may be a better choice to control viral infections.

The pathological phenomenon of Th17 cells has also been reflected in autoimmune and inflammatory diseases, including psoriasis, rheumatoid arthritis, ankylosing spondylitis, SLE and multiple sclerosis [17]. Altered levels of Th17 cells have been the biggest disease-promoting factor in psoriasis [120]. It has been documented that Th17 in the dermis of psoriatic skin elicits self-lipid antigen production and inflammation [121]. Several mAbs (secukinumab, ixekizumab, bimekizumab, brodalumab, risankizumab, trldrakizumab and guselkumab) that target IL-17, IL-23 and IL-17RA have proven efficacious in the treatment of psoriasis [122]. Similarly, Th17 cells were shown to play a crucial role in joint inflammation and cartilage damage in autoimmune arthritis [123]. Despite the role of Th17 cells in SLE and multiple sclerosis remains skeptical, the fact that it bypasses the blood–brain barrier and accumulates at lesional sites showcases its participation in disease development [124]. It has also been portrayed that Huntington’s disease is a Th17 inflammatory reaction-mediated autoimmune disorder [125]. Besides, research on the active role of Th17 cells in autoimmune uveitis and Type I diabetes is in progress where inflammation is one of the core pathologies involved in disease progression.

On the side of genetic disorders, though much attention has not been drawn to studying Th17 pathology, it has still been acknowledged that Th17 are instructive cells that impact the pathological phenotype of sickle cell disease [126]. The levels of IL-17 are also upregulated in autosomal dominant polycystic kidney disease (ADPKD) [127], whereas its definitive role is yet to be explored. Certainly, a clear picture of the pathogenesis of Th17 cells and its differentiation process can help us use exosome engineering strategies to cope with the disease’s detrimental process.

## 10. Th17 Cell-Derived Exosomes in Disease Pathologies

Little is known about the molecular basis of exosomes in the regulation of Th17 differentiation toward disease progression. Recently, a study has demonstrated that contactin-1 in exosomes derived from epithelial cell signals could activate Th17 cell responses and promote asthma pathology [128]. A similar study has reported the relay of lncRNA CRNDE-h via tumor exosomes to escalate ROR-γT expression, Th17 differentiation and the activity of IL-17 cytokines [129]. In addition, LPS-stimulated human thymic mesenchymal stromal cells release exosomes that help differentiate CD4^+^ T cells into a proinflammatory Th17 cells phenotype [130]. In preeclampsia (PE) patients, exosomes significantly increase Th17 cell numbers and exacerbate PE pathogenesis [131]. Exosomes accumulated from tumor-infiltrating cells redistribute miR-451 from cancer cells to T cells, thus prompting its differentiation to the Th17 cell phenotype [132]. Noticeably, synovial fibroblast exosomes promoted Th17 differentiation under hypoxic conditions that potentiate RA pathogenesis [50]. Despite the adverse influence of exosomes in diseases, their potential in drug delivery and treatment cannot be circumvented and is currently of great research interest.

## 11. Can Exosomes Potentiate as a Bona Fide Therapeutic Candidate against Th17 Biology?

Fortunately, the ability of exosomes derived from different cells that act as drug decoys to modulate disease response has now been reported. Remarkably, granulocytic exosomes derived from bone marrow overexpress miR-29a-3p and miR-93-5P inhibit Th17 differentiation and attenuate autoimmune arthritis [133]. On the same grounds, human bone marrow mesenchymal stem cells-derived exosomes could downregulate the Th17 population and subside periodontal inflammation via modulating YAP1/Hippo signaling pathway [134]. Yet importantly, IFN-γ-stimulated mesenchymal stem cell exosomes could boost the levels of miR-125a and miR-125b that directly interfered with STAT-3 transcription and repressed Th17 differentiation in colitis mice [135]. Also, 3D exosomes improved colitis and periodontitis via restoration of Th17/Treg balance via the miR-1246/Nfat5 axis [136]. Th17 cell responses could also be strongly nullified by olfactory ecto-mesenchymal stem cells exosome in an experimental mice model of colitis [137]. Similarly, mesenchymal stem cell-derived exosomes from the human umbilical cord (UC) could negate Th17 differentiation and treat inflammatory bowel disease [138]. UC-derived blood mesenchymal stem cell exosomes could restore miR-19B/KLF13 expression to suppress Th17 cell proportion in SLE patients [139]. Interestingly, mesenchymal stem cell exosomes derived from labial glands could inhibit Th17 differentiation, and subsequently suppress IL-17 production in the treatment of Sjogren’s syndrome [140]. Of note, dendritic cell (DC) derived exosomes block effector Th17 function and protect the liver from ischemia reperfusion [141]. Bone marrow-derived mesenchymal stem cell exosomes deliver miR-23a-3p to balance the proportion of Th17 cells and halt aplastic anemia [142]. On the highlight, gene-modified DC-derived exosomes loaded with FOXP3 results in an obvious decrease in Th17 cells and ameliorate autoimmune encephalomyelitis [143]. And, TGF-β gene-modified DCs release exosomes that have been proven more efficacious in treating Th17-mediated inflammatory bowel disease [144]. Alternatively, a study has demonstrated that heat-stressed tumor-cell-derived exosomes exhibit anti-tumor effects via biased differentiation of Th17 cells [145]. In patients with hypertrophic scaring, adipose-derived stem cell exosomes could attenuate ROR-γT expression and Th17 differentiation, thus, defining its role in disease suppression [146]. Taken together, it is conceivable and of the opinion that exosomes can be used as promising bio-shuttles against Th17-associated diseases.

To further claim exosomes to be a powerful candidate as drug delivery vehicles against Th17 cells, methods of drug loading, exosome uptake efficiency and the ease to release the payload are further extrapolated. In general, drugs can be directly loaded into the exosomes or loaded into the parental cells, which are then released into the exosomes. In spite of several challenges such as nucleic acid degradation associated with electroporation and ultrasound loading techniques, they are considered the most efficient methods for exosomal drug loading. Moreover, techniques such as thermal shock and extrusion demand high-end equipment, which increases costs. Again, the addition of saponins and transfection reagents can increase the chances of drug toxicity. To highlight, though incubation methods result in low productivity, it protects the membrane from degradation and has the advantages of low cost and high safety. Howbeit, to overcome low drug yield via the incubation method, strategies of increasing drug concentration and continuous stirring during incubation can be implied [147]. Findings have demonstrated a new strategy (late domain-ubiquitin and ESCRT vesicle trafficking technique) for the successful loading of biologically active molecules that, however [148], needs further investigation for loading drugs against Th17 cells. Once drugs are sequestered inside the exosomes, the process of uptake occurs via direct membrane fusion, pinocytosis, and endocytosis (Figure 1). The uptake efficiency can be manipulated via surface engineering methods. For instance, a previous study made use of electrostatic interactions to bind cationic lipids onto exosome surfaces to increase their uptake efficiency [19]. Another modification approach includes the attachment of cell-penetrating peptides (CPPs), in particular arginine-rich CPPs, that increase exosome internalization by micropinocytosis [149]. Also, an increase in membrane rigidity via enrichment in sphingolipids and cholesterol improves fusion efficiency between exosomes and recipient cells [19]. Post internalization, a fraction of exosomes fuses with the limiting membrane of endosomes/lysosomes to release the cargo to the cell cytosol in an acidification-dependent manner. An interesting aspect studied previously demonstrated that exosome membrane integration with connexin 43 necessitates hexametric channel formation and provides a direct route for cytoplasmic transfer of exosomal payload [150]. Further detailed investigation of these methods in relation to exosomal loading of anti-Th17 drugs and their release inside Th17 cells needs to be clarified.

## 12. Exosomes as a Budding Bio-Shuttle: A Perspective in the Landscape of Th17 Cell

Several delivery units, such as liposomes, nano emulsions, nano gels and micelles, are underway for the targeted transfer of molecules in diseases. However, it cannot be overlooked that it faces several challenges, including poor solubility, stability, loading efficiency and, last but not the least, its clearance from the reticuloendothelial system [151]. By contrast, exosomes are naturally biocompatible cargo delivery “trucks” across many biological membranes. Recently, experiments have also been conducted to delineate the source of exosomes and their isolation techniques. Immature dendritic cells, for example, are the most potent source for the isolation of therapeutic exosomes as it lacks CD86 and MHC markers, and thus, possess low immunogenicity [152]. Moreover, mesenchymal stem cells are self-renewal and are a good source of exosomes as they exhibit immunosuppressive effects [153]. Above all, the exosomes from cancer cells possess significant tetraspanin content that facilitates ligand interaction in several tissues [154]. Among several biofluids, peripheral blood is a good source of exosomes obtained from different cells [155]. Other sources of exosomes include plants and fruits that can target cancer and inflammatory diseases.

Although the scope of exosome as therapeutics has been well described, its usage in ferrying drug cargo is not much explored in Th17 biology. At the cutting edge, the rationale for its design to deliver bio-shuttles against the powerful Th17 cells is to strengthen drug efficacy, overcome drug resistance and limit off-target side effects. This section of the review highlights the possible bio-shuttles that can specifically be delivered via the use of exosomes to hamper Th17 cells. A reasonable explanation will definitely be in hand to claim these bio-shuttles as anti-IL-17 therapies.

## 13. Packaging of Small Molecules

Small molecules are selective drugs that specifically function as enzyme or receptor inhibitors and allosteric modulators in order to disrupt protein–protein interactions. These drugs are prescribed with numerous warnings and advice to discontinue if the side effects are strong. Moreover, natural compounds that are small molecules show limited efficacy because of their low bioavailability. These limitations might prompt one to think that exosomes can be a prospective delivery platform for small molecules to dynamically increase their uptake, improve their availability at target sites and exhibit superior therapeutic indices. No reports have yet studied the application of exosomes in improving the delivery and efficacy of small-molecule drugs against the Th17 catastrophe. In light of these shortcomings, we list here the small molecules cargo that can be packaged into the exosomes for specific targeting of Th17 differentiation and function (Table 3).

## 14. Packaging of Proteins

Several investigations have made use of exosomes in the distribution of large molecules like peptides and proteins to treat diseases. The high potency, low toxicity and high selectivity make them outstanding therapeutic candidates over the small molecule. Howbeit, rapid clearance from the body, poor membrane permeability and poor metabolic stability demand a delivery vehicle [169]. Exosomes have been proven as a better vehicle for autoimmune diseases such as SLE, where tolerance therapy using the autoantigenic peptide, for instance, histone peptide H4_71–94_, is a highly opted therapeutic option [170]. In another instance, exosome can be a carrier for a peptide derived from the globular C-terminal domain of adiponectin (e.g., KS23) that has characteristic anti-inflammatory properties and decreases the proportion of Th17 cells as reported in experimental autoimmune uveitis [171]. Interestingly, packaging antagonistic peptides that target the first and second extracellular loops of chemokine CCR5 and p19 subunit of IL-23 into exosomes may be effective in suppressing the IL-23/Th17 axis [172]. As αv integrin is present in dendritic cell populations, it necessitates Th17 development, whereas an antagonist peptide, VnP-16, when packaged inside exosomes and administered, could result in impairing Th17 differentiation and exhibit better clinical outcomes [173]. Notwithstanding, it is tempting to speculate that N-formyl peptide receptor 2 (FPR2) agonists, such as scolopendrine, Cpd43 and WKYMVm, could be loaded into exosomes to suppress dendritic cell maturation and inhibit Th17 responses, as demonstrated in a mouse model of autoimmune arthritis [174]. Of note, an exosome-released peptide agonist to ADAM17 that is responsible for membrane shedding of IL-23R can block Th17 functions in several Th17-associated diseases [175]. Critically, an antagonist peptide as a drug delivered by exosomes to CD71 can disrupt Th17 differentiation [176]. Furthermore, exosome delivery of antagonistic peptides, which target miRNA (e.g., miR-21), that incite Th17 pathogenesis can be a strategy for a peptide-based therapeutic approach.

## 15. Packaging of Genetic Substances

As exosomes are efficient in preventing nucleic acids (siRNAs and miRNAs) from degradation, they represent ideal vectors for gene therapy approaches. Besides, exosomes are naturally produced transporters of RNAs intercellularly. In our perspective, Phosphatase and tensin homolog (PTEN) regulatory miR-22 can be loaded into exosomes to repress Th17 cell responses [177]. In another instance, exosome-based delivery of miR-221-5p could be used to efficiently target the suppressor of cytokine signaling (SOCS)1 to inhibit Th17 cells in asthma [178]. On the upper hand, exosome-based delivery of miR-340 might be capable of attenuating psoriasis by directly targeting IL-17A [179]. In addition, nuclear receptor 4A2 (NR4A2) specific siRNA can be used as a drug to be delivered by exosomes and decrease Th17 effector function [180].

More recently, the introduction of CRISPR/Cas9 technology has revolutionized the treatment of several diseases. It has been an attractive approach due to the high precision and flexibility of manipulating a specific gene. However, finding a suitable vehicle to deliver CRISPR/Cas9 is a major hurdle [181]. Exosomes, as natural vehicles, possess low immunogenicity and can therefore be a potent approach in delivering CRISPR/Cas9 system into cells. It is true that exosomes can be exploited for CRISPR-mediated blockade of glycolysis, which can result in the loss of Th17 cells [182]. As a key point, targeting oxidative phosphorylation via exosome-based delivery of CRISPR/Cas9 might abrogate Th17 cells’ resistance to apoptosis [183], which can be a promising approach. It is probable that CRISPR/Cas9 release via exosomes to manipulate RNA binding proteins, such as HuR, can also help protect against Th17 responses [113,184].

## 16. Strategies to Engineer Exosomes for Specific Targeting of Th17 Cells

Nowadays, therapeutic exosomes are delivered to specific sites via active targeting strategies that utilize techniques to modulate the surface of exosomes. There are two ways for active targeting of exosomes, a non-genetic approach that defines direct manipulation of exosomal surface and genetic approaches that indirectly modulates the exosomes via genetic manipulation of its cellular source [29]. Here, we brief on the exosomal surface engineering that can be successfully accomplished for the targeted delivery of exosomes to Th17 cells.

The non-genetic/direct method utilizing chemical (click chemistry) or physical techniques (hydrophobic insertion and receptor–ligand interaction) can help acquire targetability in therapeutic exosomes. Click chemistry is a covalent bonding strategy in which the ligand molecules form covalent bonds with exosomes [185]. For instance, designed peptides for Th17 cell-specific receptors or integrins can be integrated into exosomes via click chemistry to generate Th17 cell-targeting exosomes [186]. It is a matter of debate whether α_v_β_3_ targeted peptide can be conjugated to exosome surface via click chemistry to target Th17 cells. However, the method of click chemistry can be detrimental as it necessitates toxic chemicals for stabilizing bonds, thus, alarming for its use in therapeutics. Apart from the covalent binding method, physical methods can opt as one of the options for allowing the binding of ligand moieties to exosome membranes. Attachment of CD30 aptamer to exosomes by hydrophobic interactions can be employed for Th17 pathological condition as CD30 is actively expressed by activated T cells [187] (Figure 3).

Further, genetically engineered exosomes warrant attention in the active targeting of Th17 cells. To deliver exosomes carrying therapeutics to Th17 cells, targeting entities, such as antibodies or peptides that could bind specifically to integrins or receptors expressed on Th17 cells, should be showcased on the exosome surface through a genetic modification approach. Exosomes expressing anti-prohibitins attached to GPI-anchored proteins or the C1C2 domain of cadherin in exosomes can be targeted to Th17 cells, as prohibitins are highly expressed on the surface of murine and human Th17 cells [188]. In another instance, exosomes can be genetically manipulated to express a disintegrin and metalloproteinase 15 that can bind to α_v_β_3_ overexpressing Th17 cells [189]. Overall, it is undeniable that efforts to express Th17-targeting entities on the surface of exosomes via conjugation with the GP1 and C1C2 domains and tetraspanins associated with exosome membranes can be a promising approach for the active targeting of the Th17 cells.

## 17. Present Challenges and Future Perspectives

Numerous exosomes are filed for clinical trials to assess the risk-benefit ratio, and their discovery as an outstanding drug delivery platform is opening the door toward the cure of several diseases. Unfortunately, cells produce a small number of exosomes that are insufficient for clinical studies, and miRNA that is loaded as therapeutic cargo, if anionic, fails to cross biological membranes and accumulates in kidneys and livers due to its short half-life. Specifically, little knowledge of the targeting entities of Th17 cells shields the exploration of exosomes against their pathogenicity. Despite numerous challenges, such as inadequate characterization methods, a lack of specific markers and stability issues, these naturally derived vesicles have tremendous potential in biomedical research. Therefore, it is suggestible to find ways to overcome the shortcomings and obstacles to label exosome drug delivery as a safe and efficient approach in clinics.

Several leading companies across the globe are conscious of overcoming the scientific and technological challenges faced during exosome clinical trials. As an alternative, synthetic exosome mimics or cell membrane-cloaked nanoparticles have been fabricated for drug delivery with the potency to increase drug half-life while still sustaining equal biodistribution and clearance rates as that of natural exosomes. The development of cost-effective strategies and isolation and exosome characterization methods with high sensitivity for large-scale production is in progress. Exciting new approaches for effective drug loading are being explored. Certainly, this will position exosomes as an attractive “waste disposal unit” that can be exploited for therapeutic usage in clinics.

In conclusion, it is envisaged that the integration of reliable isolation methods and drug packaging techniques will gear up its validation in in vivo studies that may lay a solid foundation to solve the puzzle of its clinical translation.

## Figures and Tables

**Figure 1 ijms-24-07647-f001:**
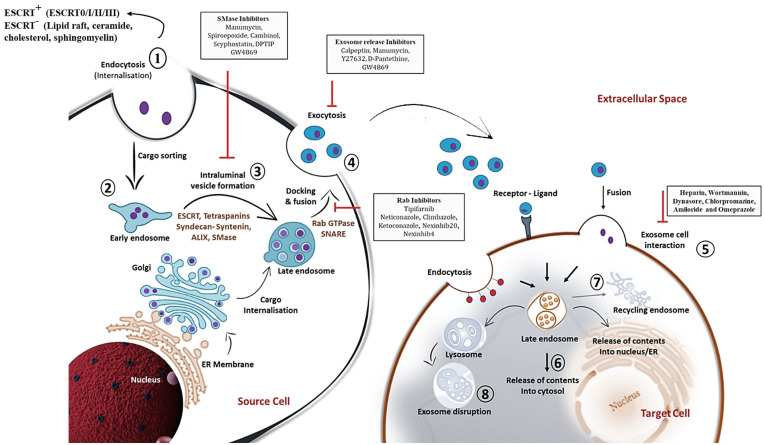
Detailed schematic overview of exosome biogenesis and inhibitors that block the release of exosomes within the endosomal system. The formation of exosomes involves the inward budding and fusion of the limiting membrane of endocytic vesicles that engender intraluminal vesicles (ILVs). The maturation process of ILVs can be through endosomal complexes required for transport (ESCRT)-dependent and -independent mechanisms that process cargo sorting and formation of multivesicular bodies (MVBs). Components of MVBs can then be integrated into the membranes for release into the extracellular space or may be guided for lysosomal degradation. Exosomal contents released into the extracellular space can then be internalized by the target cell via membrane fusion, ligand-receptor conformation or endocytosis mechanism. This illustration further comprehends various chemical inhibitors, including SMase inhibitors, exosome release inhibitors and Rab inhibitors (depicted in black rectangular boxes) that prevent exosome biogenesis and secretion within the endosomal compartment. The sequential steps on exosome biogenesis follows: 1. Exosome formation; 2. Cargo sorting; 3. MVBs formation; 4. Exosome release; 5. Exosome-target cell interaction; 6. Cargo release into cytosol of target cells; 7. MVBs endosome recycling; and 8. Lysosome degradation of exosomes. Abbreviations: Rab; Ras associated binding protein, SNARE; SNAP receptor.

**Figure 2 ijms-24-07647-f002:**
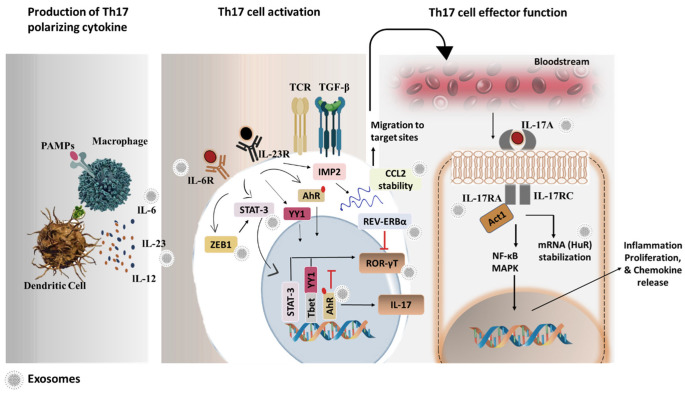
Pictorial representation of exosomal targets that participate during the T helper (Th)17 cell developmental stages and in promoting its effector function. Early activation of antigen-presenting cells (APCs) mounts an immune response that releases Th17 cell polarizing cytokines, such as IL-6, IL-23, IL-12, etc. These cytokines interact with the receptor on CD4^+^ T cells and induce its polarization to the Th17 cell phenotype via activation of intermediate kinases and transcription factors. Differentiated Th17 cells then migrate via the bloodstream to inflammatory sites and execute effector function. Furthermore, the diagram highlights multi-targets, including cytokines or its receptor, transcriptional factors or RNA-binding proteins that can be modulated via exosomal anti-Th17 therapies. Abbreviations: PAMPs; pathogen-associated molecular pattern molecules, STAT-3; signal transducer and activator of transcription 3, ZEB1; zinc finger E-box binding homeobox 1, AhR; aryl hydrocarbon receptor, YY1; yin yang 1, IMP2; IGF2 mRNA binding protein 2, TGF-β; transforming growth factor β, TCR; T cell receptor, CCL2; chemokine (C-C motif) ligand 2, REV-ERBα; nuclear receptor subfamily 1group D, ROR-γT; retinoic acid-related orphan receptor gamma t, HuR; human antigen R.

**Figure 3 ijms-24-07647-f003:**
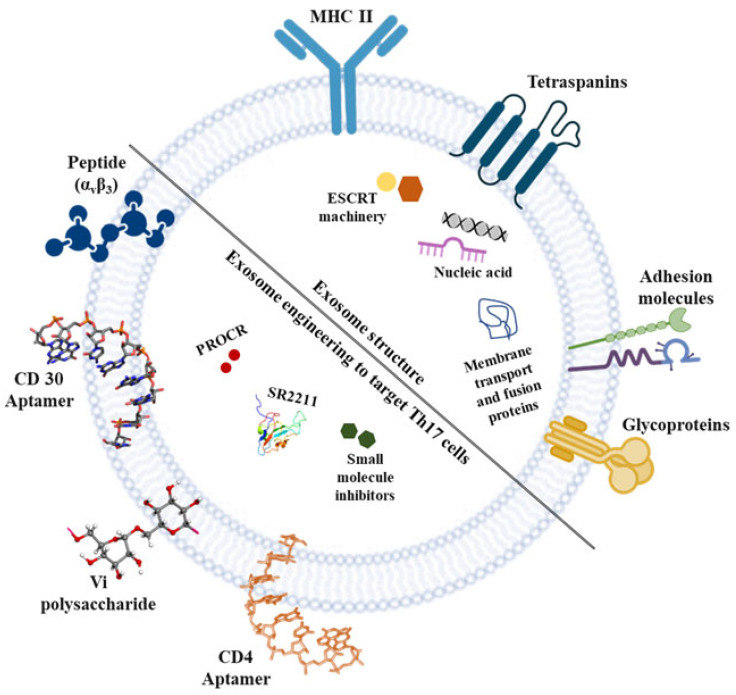
Diagrammatic illustration of exosome engineering for loading molecules into the lumen or sorting them on their surface for specific targeting of Th17 cells. Exosomes are engineered to express CD30 or CD4 aptamer, peptide targeting α_v_β_3_ and Vi polysaccharide (targets prohibitins) on their surface using different sorting modules that result in a sustained response against Th17 cells. Furthermore, the packaging substances including drugs, nucleic acid molecules, etc., can be loaded onto the engineered exosomes for targeting Th17 cells to inhibit its pathogenic mechanism in diseases. Abbreviations: PROCR; protein C receptor, MHC; major histocompatibility complex.

**Table 1 ijms-24-07647-t001:** List of exosomal RNAs in the occurrence of disease.

Exosome Source	Exosome Cargo	Disease	Disease Outcome	Reference
HCT116 and Serum	miR-25, miR-130b and miR-425	Colorectal Cancer	Aggravates liver metastasis	[52]
Serum	miR-1247-3p	Liver cancer	Promotes lung metastasis	[52]
A2780 CCM	miR-223	Epithelial ovarian cancer	Promotes chemoresistance	[52]
Variable	miR-21	Multiple cancers	Promotes cancer	[52]
Serum	miR-7977	Lung adenocarcinoma	Increase in proliferation, invasion and inhibits apoptosis	[52]
Pan02 CCM	miR-155-5p and miR-221-5p	Pancreatic ductal adenocarcinoma	Promotes metastasis	[52]
HT-29/SW480	miR-375-3p	Colon cancer	Induces EMT	[52]
MSC	miR-21-5p	Breast cancer	Promotes chemoresistance	[52]
Plasma	miR-1-3p	Sepsis	Endothelial cell dysfunction	[52]
Serum	miR-4449	Diabetic kidney	Promotes pro-inflammation & oxidative stress	[53]
MGC803, MKN45, HGC27, and SGC7901 CCM	miR-21-5p	Gastric cancer	Promotes peritoneal metastasis	[54]
Human bronchial epithelial cells	miR-21 and miR-210	COPD	Promotes myofibroblast differentiation and hypoxia	[55]
Serum	miR-96, miR-222-3p, miR-499a-5p	Lung cancer	Promote cell migration and invasion	[55]
Induced pluripotent stem cells (IPSC)-derived astrocytes/microglia	miR-21-5p	Adenocarcinoma	Induce neurotoxic reactive astrocytosis, cognitive impairment	[56]
TDEs	miR-141	Lung cancer	Induces angiogenesis and malignancy	[57]
TDEs	miR-107	Gastric Cancer	Promote immunosuppression	[58]
SCLC	miR-375-3p	Lung Cancer	Disrupts vascular barrier	[59]
Urine	miR-200b	Renal fibrosis	Fibrosis progression	[60]
Plasma	lnc-MKRN2	Parkinson Disease	Develops disease occurrence	[52]
Serum	lncRNA-UCA1	Pancreatic Cancer	Promotes angiogenesis	[52]
Serum	HOXD-AS1	Prostate Cancer	Promotes metastasis	[52]
Urine	lncBCYRN1, lncLNMAT2	Bladder Cancer	Promotes lymphatic metastasis	[52]
CCM	LncPCGEM1	Gastric cancer	Induces metastasis and migration	[52]
TGF-β A549	lnc-MMP2-2	Lung cancer	Promotes invasion and metastasis	[55]
A172 cells	POU3F3	Glioma	Promotes angiogenesis	[61]
CAF	MEG3	SCLC	Cisplatin resistance	[62]
MCF7	MALAT1	Breast cancer	Promotes proliferation	[63]
Plasma	circ-RanGAP1	Gastric cancer	Promotes metastasis	[52]
HCC CCM	circ-RNA-100338	HCC	Promotes angiogenesis and invasion	[52]
Serum	Circ-0006156	Thyroid cancer	Promotes tumorigenesis	[52]
TDEs	PDE8A	Pancreatic cancer	Elevates invasive growth	[64]
L-02 CCM	circ-100284	Hepatocarcinoma	Accelerates cell cycle and proliferation	[65]

Abbreviations: HCT, Human colorectal cancer cell line; CCM, Cell culture media; EMT, Epithelial to mesenchymal transition; MSC, Mesenchymal stem cell; COPD, Chronic obstructive pulmonary disease; TDEs, Tumor-derived exosomes; SCLC, Small cell lung cancer; CAF, Cancer-associated fibroblasts; MALAT1, Metastasis-associated lung adenocarcinoma transcript 1.

**Table 2 ijms-24-07647-t002:** Summary of exosomal cargo in pre-clinical and clinical trials as therapeutics.

Exosome Source	Exosome Cargo	Disease	Clinical Status	Reference
Plant (Grapes)	Curcumin	Colon cancer	NCT01294072 Phase I	-
Plant (Ginger)	Curcumin	Inflammatory Bowel Disease	NCT04879810 (Completed)	-
Dendritic cell	Dex2	Non-small cell lung cancer	NCT01159288 (Completed)	-
Plant (Grapes)	Lortab	Oral mucositis	NCT01668849	-
Mesenchymal stromal cells	KRASG12D siRNA	Pancreatic Ductal Adenocarcinoma	NCT03608631 (Phase I)	-
Blood	Anlotinib	Non-small cell lung cancer	NCT05218759	-
Blood	Pembrolizumab	Head and neck cancer	NCT04453046 (Terminated)	-
Dendritic cell/macrophage	Chimeric exosomal tumor vaccines	Bladder cancer	NCT05559177 (Phase I)	-
Circulating lymphocytes and serum	Merck 3475 Pembrolizumab	Triple-negative breast cancer	NCT02977468 (Phase I)	-
Liquid biopsies	18F-DCFPyL PET/CT	Prostatic neoplasms	NCT03824275 (Phase II/III)	-
Macrophage	CDK-004	Gastric cancer, colorectal cancer	NCT05375604 (Phase I)	-
Human cell	miR-497	Lung cancer	-	[79]
Dental pulp stem cell	Chitosan hydrogel	Experimental periodontitis	-	[80]
MSC-NTF	-	COVID-19-induced ARDS	-	[81]
LX-2 cells	Cas9 ribonucleoprotein	Liver diseases	-	[82]

**Table 3 ijms-24-07647-t003:** List of drugs that can be exosomal cargo for specific targeting of Th17 cells.

Drug	Structure	Molecular Target	Reference
Digoxin	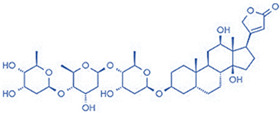	ROR-γT	[156]
JNJ-61803534	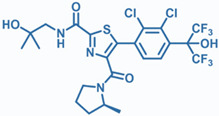	ROR-γT	[157]
BMS-986251	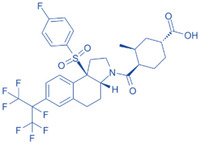	ROR-γT	[158]
Izumerogant (IMU)-935	-	ROR-γT	Phase I clinical trial
MS402	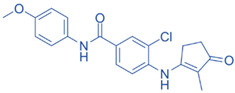	BET N-terminalbromodomain	[159]
Zebularine	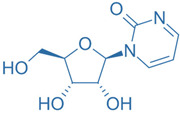	IL-17	[160]
Birabresib (OTX015)	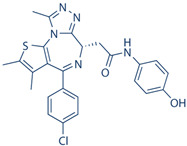	BET inhibitor	[161]
MDL-101	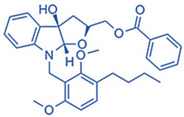	IL-6	[162]
Tapinarof	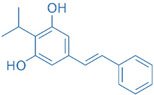	AhR	[163]
LMT-28	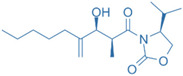	IL-6/GP130, STAT-3	[164]
Neobaicalein	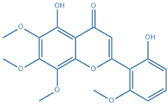	STAT-3	[165]
Bazedoxifene	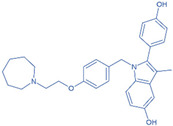	IL-6/GP130	[166]
2,20,40-Trihydroxychalcone	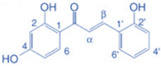	ROR-γT	[167]
Ursolic Acid	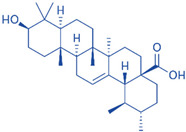	ROR-γT	[166]
IOX1	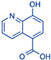	TET2	[168]

## Data Availability

Not applicable.

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
