# Peer review of "Leveraging Exosomes as the Next-Generation Bio-Shuttles: The Next Biggest Approach against Th17 Cell Catastrophe"

_ijms, 2023, doi:10.3390/ijms24087647_

Round 1

Reviewer 1 Report

The current state of this review paper is up to the standard of Journal of Molecular Sciences and could be of an interest to a wide range of readers of this journal, however, there are a few suggestion, kindly find them below, to further improve this review survey.

This literature survey by Samarpita et al on the use of exosomes as a means for cargo delivery is relevant and addresses a specific gap in the field, as it discusses the potential of exosomes as drug carriers for targeting Th17 cells. The review is well-written and organized, but would benefit from an English and structural revision.

The main question addressed by the research is whether exosomes can be used as a promising approach for targeted drug delivery against Th17 cells in diseases, given the current limitations of targeted approaches such as high cost of production, poor bioavailability, rapid transformation, and opportunistic infections.

The topic is relevant in the field and addresses a specific gap by discussing the potential of using exosomes as drug carriers for targeting Th17 cells, which could overcome the limitations of current targeted approaches.

The review adds to the subject area by providing a comprehensive overview of exosome biogenesis, current clinical trials, and challenges of exosomes as drug carriers, with a focus on their potential use in targeting Th17 cells. The review also discusses the future prospects of exosome bioengineering for targeted drug delivery against Th17 cells.

As this is a review paper, the methodology is not applicable. However, the authors could have included more information about the sources of the studies they cited, such as the sample size, study design, and the statistical methods used to analyze the data.

Additionally, it would be helpful if the authors included a section on the current state of the art in the classification of exosomes.

The conclusions are consistent with the evidence and arguments presented and address the main question posed in the abstract.

The references are appropriate and relevant to the topic discussed. However, to further strengthen the references the authors are suggested to add more references on the current methods of exosomes’ classification (e.g., doi.org/10.1002/adsr.202200039 doi.org/10.1002/adma.202211602)

Author Response

Reviewer 1

The current state of this review paper is up to the standard of the Journal of Molecular Sciences and could be of an interest to a wide range of readers of this journal, however, there are a few suggestions, kindly find them below to further improve this review survey.

  1. Comment: This literature survey by Samarpita et al on the use of exosomes as a means for cargo delivery is relevant and addresses a specific gap in the field, as it discusses the potential of exosomes as drug carriers for targeting Th17 cells. The review is well-written and organized but would benefit from an English and structural revision.

Response: We thank the reviewer for acknowledging the specific gap in the field that has been addressed in this review. As per the suggestion, the review has now been scrutinized for English corrections by a language expert.

  1. Comment: The main question addressed by the research is whether exosomes can be used as a promising approach for targeted drug delivery against Th17 cells in diseases, given the current limitations of targeted approaches such as high cost of production, poor bioavailability, rapid transformation, and opportunistic infections.

Response: The authors thank the reviewer for the in-depth understanding of the review.

  1. Comment: The topic is relevant in the field and addresses a specific gap by discussing the potential of using exosomes as drug carriers for targeting Th17 cells, which could overcome the limitations of current targeted approaches.

Response: Thank you for the valuable comment.            

  1. Comment: The review adds to the subject area by providing a comprehensive overview of exosome biogenesis, current clinical trials, and challenges of exosomes as drug carriers, with a focus on their potential use in targeting Th17 cells. The review also discusses the prospects of exosome bioengineering for targeted drug delivery against Th17 cells.

Response: We thank the reviewer for an in-depth comment. The review was in purpose of tackling the current problems associated with the liposomes and make the reader understand the value of using exosomes as drug delivery vehicles to modulate the immune system, the important strategy followed to combat most of the diseases.

  1. Comment: As this is a review paper, the methodology is not applicable. However, the authors could have included more information about the sources of the studies they cited, such as the sample size, study design, and the statistical methods used to analyze the data.

Response: As per the suggestion, the authors have now incorporated the sample size, wherever applicable, in the revised manuscript. However, the sample size and study design pertaining to clinical trials can be referred from the identifier mentioned in the table.

  1. Comment: Additionally, it would be helpful if the authors included a section on the current state of the art in the classification of exosomes.

Response: We thank the reviewer for the suggestion. The authors have now included the information pertaining to the classification of exosomes in the revised manuscript.

  1. Comment: The conclusions are consistent with the evidence and arguments presented and address the main question posed in the abstract.

Response: Thank you for the valuable comment.

  1. Comment: The references are appropriate and relevant to the topic discussed. However, to further strengthen the references the authors are suggested to add more references on the current methods of exosomes’ classification (e.g., doi.org/10.1002/adsr.202200039 doi.org/10.1002/adma.202211602).

Response: As suggested, the references have now been included in the revised manuscript in the section with “exosome classification and its bio-distribution: the current state of the art”.

Reviewer 1

The current state of this review paper is up to the standard of Journal of Molecular Sciences and could be of an interest to a wide range of readers of this journal, however, there are a few suggestions, kindly find them below to further improve this review survey.

  1. Comment: This literature survey by Samarpita et al on the use of exosomes as a means for cargo delivery is relevant and addresses a specific gap in the field, as it discusses the potential of exosomes as drug carriers for targeting Th17 cells. The review is well-written and organized but would benefit from an English and structural revision.

Response: We thank the reviewer for acknowledging the specific gap in the field that has been addressed in this review. As per the suggestion, the review has now been scrutinized for English corrections by a language expert.

  1. Comment: The main question addressed by the research is whether exosomes can be used as a promising approach for targeted drug delivery against Th17 cells in diseases, given the current limitations of targeted approaches such as high cost of production, poor bioavailability, rapid transformation, and opportunistic infections.

Response: The authors thank the reviewer for the in-depth understanding of the review.

  1. Comment: The topic is relevant in the field and addresses a specific gap by discussing the potential of using exosomes as drug carriers for targeting Th17 cells, which could overcome the limitations of current targeted approaches.

Response: Thank you for the valuable comment.            

  1. Comment: The review adds to the subject area by providing a comprehensive overview of exosome biogenesis, current clinical trials, and challenges of exosomes as drug carriers, with a focus on their potential use in targeting Th17 cells. The review also discusses the prospects of exosome bioengineering for targeted drug delivery against Th17 cells.

Response: We thank the reviewer for an in-depth comment. The review was in purpose of tackling the current problems associated with the liposomes and make the reader understand the value of using exosomes as drug delivery vehicles to modulate the immune system, the important strategy followed to combat most of the diseases.

  1. Comment: As this is a review paper, the methodology is not applicable. However, the authors could have included more information about the sources of the studies they cited, such as the sample size, study design, and the statistical methods used to analyze the data.

Response: As per the suggestion, the authors have now incorporated the sample size, wherever applicable, in the revised manuscript. However, the sample size and study design pertaining to clinical trials can be referred from the identifier mentioned in the table.

  1. Comment: Additionally, it would be helpful if the authors included a section on the current state of the art in the classification of exosomes.

Response: We thank the reviewer for the suggestion. The authors have now included the information pertaining to the classification of exosomes in the revised manuscript.

  1. Comment: The conclusions are consistent with the evidence and arguments presented and address the main question posed in the abstract.

Response: Thank you for the valuable comment.

  1. Comment: The references are appropriate and relevant to the topic discussed. However, to further strengthen the references the authors are suggested to add more references on the current methods of exosomes’ classification (e.g., doi.org/10.1002/adsr.202200039 doi.org/10.1002/adma.202211602).

Response: As suggested, the references have now been included in the revised manuscript in the section with “exosome classification and its bio-distribution: the current state of the art”.

Reviewer 2 Report

In this review, the authors shed light on the usage of exosomes to hamper T helper 17 cells (Th17) including strategies to engineer exosomes for specific targeting of Th17 cells. T-cell mediated immune responses need to be regulated to avoid the development of autoimmune or chronic inflammatory diseases. This is a comprehensive well organized review with a pleasant flow which take into account also the structure and composition of exosomes, their biogenesis and release as well as their properties to be used as drug delivery system for cancer and other related diseases taking into account pre-clinical and clinical trials. While I am generally enthusiastic about the topics described including the clarity of the scope of this review, the article requires further refinement in the content of the information presented.

Here are my detailed comments:
1. What are the percentage of uptake of therapeutic exosomes? Are there exosomes that are better uptaken rather than others?

2. Does their biodistribution vary according to source cell type and delivery approach?

3. What is the yield of the drug loaded in relation of the different drug loading methods?

4. Does the release of the payload change depending on the cell source? Which methods are usually employed to release the payloads?

5. How do patients respond to therapeutic exosomes?

6. How is the stability of the exosomes as drug delivery system over time? Does the inner content remain functional for long time?   

7. In the section “challenges and future perspectives” the authors claim that to overcome the small amount of exosomes produced by the cells, synthetic exosomes mimics have been fabricated for drug delivery. Are any differences in therapeutic efficiency between native and artificial exosomes? What about biodistribution and clearance?

Author Response

Reviewer 2

In this review, the authors shed light on the usage of exosomes to hamper T helper 17 cells (Th17) including strategies to engineer exosomes for specific targeting of Th17 cells. T-cell mediated immune responses need to be regulated to avoid the development of autoimmune or chronic inflammatory diseases. This is a comprehensive well-organized review with a pleasant flow which take into account also the structure and composition of exosomes, their biogenesis and release as well as their properties to be used as drug delivery system for cancer and other related diseases taking into account pre-clinical and clinical trials. While I am generally enthusiastic about the topics described including the clarity of the scope of this review, the article requires further refinement in the content of the information presented.

Here are my detailed comments:

  1. Comment: What are the percentage of uptake of therapeutic exosomes? Are there exosomes that are better uptaken rather than others?

Response: We thank the reviewer for this useful comment. It should be noted that though there are little information available regarding the uptake efficiency of exosomes, however, it is understood that cells uptake exosomes via various endocytic pathways as described in Fig. 1 of the manuscript. Previous reports have demonstrated that exosomes with cargo are more efficient than cargo itself that remained below 1% post 24 hours [Brien et al., 2022]. Again, the uptake efficiency was high for paclitaxel-loaded macrophage-derived exosomes in comparison to paclitaxel-loaded liposomes in the 3LL-M227 murine Lewis lung carcinoma cell line. Further uptake efficiency is validated via fluorescence microscopy and flow cytometry. No concrete data exist regarding the percentage of uptake; however, it has been demonstrated that efficiency varies from range 11.00 – 48.00 % [Huyan et al., 2018]. Based on this evidence, it can thus be concluded that the uptake of exosomes is based on the parental cell from which they are derived though the difference may not be significant [Huyan et al., 2018].

With reference to exosome varieties and its uptake efficiency, exosome surface bioengineering can increase the uptake efficiency of exosomes. For example, a previous study has made use of electrostatic interactions to bind cationic lipids on the surface of exosomes to increase its uptake efficiency [Luan et al., 2017]. Again, blood derived exosomes have been modified with transferrin-conjugated superparamagnetic nanoparticles targeting the native transferrin receptors present on the exosome membrane that increase its uptake into the recipient cells [Luan et al., 2017].

  1. Comment: Does their biodistribution vary according to source cell type and delivery approach?

Response: Thank you for the comment. Regardless of the origin, it has been stated that the biodistribution of exosomes are largely to liver, spleen, kidney and lungs. However, there are evidences that state that exosome biodistribution varies according to parental cell and the route of administration. Like, glioma-derived exosomes offer an effective therapeutic strategy for drug delivery to the brain. Again, intramuscular administration of exosomes increases it bio-availability in the gastrointestinal tract. Alternatively, the oral gavage method of administration localized it more in the intestine [Kang et al., 2021]. Besides the route of administration and parental cell, biodistribution can be altered via surface modifications of exosomes using direct or indirect methods. Exosome were modified with A33 antibody for targeted delivery of doxorubicin against colorectal cancer [ Li et al., 2018]. Other instances of exosome bioengineering for targeted delivery have been mentioned in the review under section “strategies to engineer exosomes for specific targeting of Th17 cells”.

  1. Comment: What is the yield of the drug loaded in relation of the different drug loading methods?

Response: We thank the reviewer for the query. Please note, drugs can be directly loaded into the exosomes or loaded into the parental cells which are then released into the exosomes. Despite several advantages and disadvantages associated with loading techniques, electroporation and ultrasound methods are most efficient methods of drug loading. However, electroporation can lead to nucleic acid degradation and ultrasound can alter surface of exosomes. Moreover, methods like thermal shock and extrusion demands high equipment’s and high costs. Again, addition of saponins and transfection reagents can increase drug toxicity. To highlight, though incubation methods result in low yield, it does not destroy the membrane and have advantages of low cost and high safety. To overcome low drug yield via incubation method, strategies of increasing drug content and stirring during incubation can be implied [Xi et al., 2021].  

  1. Comment: Does the release of the payload change depending on the cell source? Which methods are usually employed to release the payloads?

Response: Thank you for the query. In general, cells can take up exosomes by various means including direct membrane fusion, pinocytosis, and endocytosis. And then, a fraction of internalized exosomes fuses with the limiting membrane of endosomes/lysosomes in an acidification dependent manner that results in release of cargo to the cell cytosol. However, several methods can be followed to facilitate entry of exosomes into cells. One approach is to equip exosomes with cell-penetrating peptide (CPPs), such as arginine-rich CPPs, that enhances exosome internalization by micropinocytosis. Again, increasing membrane rigidity by enriching sphingolipid and cholesterol improves fusion efficiency between exosomes and the recipient cells. Moreover, integration of exosomes with connexin 43 (Cx43), a membrane protein that assembles to form hexametric channels provides another route for direct cytoplasmic transfer of exosome payload (Shimaoka et al., 2019).

  1. Comment: How do patients respond to therapeutic exosomes?

Response: We thank the reviewer for the query. Based on the number of ongoing clinical trials and efforts to spur exosomes as drug vehicles in clinics indicates its high efficiency in patients. (https://clinicaltrials.gov/ct2/home). However, few are completed (NCT04879810) and most of the trials are on-going or at phase III clinical trials that depicts its potency as delivery vehicles.

  1. Comment: How is the stability of the exosomes as drug delivery system over time? Does the inner content remain functional for long time?

Response: Thank you for the comment. Exosomes are highly biocompatible and stable due to it ability to resist degradation by the activity of digestive enzymes until they reach the target organs [Akuma et al., 2019].

With reference to the inner content of exosomes, it has properties of increasing drug half-life and drug stability as it protects it from pH variation and extracellular proteases. For instance, the main advantage of milk-derived exosomes is that they enable the effective delivery of encapsulated therapeutic molecules through the oral cavity due to its stability under low pH-degrading gastric conditions [Kim et al., 2021]. Furthermore, drugs with low stability like curcumin are encapsulated inside the exosomes to enhance its stability and sustained release thus, highlighting the importance of exosomes to maintain the drug stability for longer time and its prospects as drug delivery vehicles in clinics.

  1. Comment: In the section “challenges and future perspectives” the authors claim that to overcome the small amount of exosomes produced by the cells, synthetic exosomes mimics have been fabricated for drug delivery. Are any differences in therapeutic efficiency between native and artificial exosomes? What about biodistribution and clearance?

Response: We thank the reviewer for the query. The authors would like to inform that exosome-mimics showed similarities to exosomes in their zeta-potential, size distribution, and morphology and possessed the immunocompatibility and stability of exosomes due to coating of the plasma membrane. Based on these perspective, exosomes or exosomes mimics unless bio-engineered have similar biodistribution and clearance rates in the biological systems. However, exosomes mimics only vary from exosomes during cargo loading as it cannot include specific mechanisms, such as the ESCRT-dependent and lipid raft-dependent pathways and have to depend on direct incubation, physical and chemical methods.
